# Cancer-Associated Fibroblasts Promote Tumor Aggressiveness in Head and Neck Cancer through Chemokine Ligand 11 and C-C Motif Chemokine Receptor 3 Signaling Circuit

**DOI:** 10.3390/cancers14133141

**Published:** 2022-06-27

**Authors:** Wen-Yen Huang, Yaoh-Shiang Lin, Yu-Chun Lin, Shin Nieh, Yi-Ming Chang, Tsai-Yu Lee, Su-Feng Chen, Kuender D. Yang

**Affiliations:** 1Institute of Clinical Medicine, National Yang Ming Chiao Tung University, Taipei 112, Taiwan; hwyyi@yahoo.com.tw; 2Department of Radiation Oncology, Tri-Service General Hospital, National Defense Medical Center, Taipei 114, Taiwan; 3Department of Otorhinolaryngology, Head and Neck Surgery, Kaohsiung Veterans General Hospital, Kaohsiung 813, Taiwan; yaohshiang@gmail.com; 4Department of Pathology, Tri-Service General Hospital, National Defense Medical Center, Taipei 114, Taiwan; yuchunlin.md@hotmail.com (Y.-C.L.); niehshin1014@yahoo.com.tw (S.N.); yiminnn1491@gmail.com (Y.-M.C.); 5Department of Pathology, Kaohsiung Veterans General Hospital, Kaohsiung 813, Taiwan; 6Division of Colon and Rectum Surgery, Department of Surgery, Tri-Service General Hospital Songshan Branch, National Defense Medical Center, Taipei 105, Taiwan; leechoumail@gmail.com; 7Department of Dentistry, School of Dentistry, China Medical University, Taichung 406, Taiwan; 8Division of Medical Research, MacKay Children’s Hospital, Taipei 104, Taiwan; 9Department of Immunology & Microbiology, National Defense Medical Center, Taipei 114, Taiwan

**Keywords:** head and neck cancer, cancer-associated fibroblasts, CCL11, CCR3, tumor microenvironment

## Abstract

**Simple Summary:**

Certain tumor aggressiveness-associated mediators from cancer-associated fibroblasts (CAFs) in tumor microenvironment have been reported. Using gene expression analysis, we identified that CAFs overexpress Chemokine ligand 11 (CCL11), which is associated with tumor migration and invasion, increased expression of cancer stem cell properties, and induction of the epithelial-to-mesenchymal transition. Neutralization of CAF-induced CCL11 reversed the aggressive phenotype of cancer cells. Based on the immunohistochemical staining of clinical samples, we found that increased co-expression of CCL11 and its receptor, C-C Motif Chemokine Receptor 3 (CCR3), was associated with poor overall survival. Our results suggest that targeting CCL11-CCR3 signaling is a potential therapeutic strategy for patients with aggressive head and neck cancer.

**Abstract:**

The tumor microenvironment (TME) plays a crucial role in tumor progression. One of its key stromal components, cancer-associated fibroblasts (CAFs), may crosstalk with cancer cells by secreting certain cytokines or chemokines. However, which important mediator(s) are released by CAFs, and the underlying molecular mechanism, remain largely unknown. In the present study, we isolated patient-derived CAFs and normal fibroblasts (NFs). Using microarray analysis, we detected chemokine ligand 11 (CCL11) overexpression in CAFs compared to NFs. CCL11 administration promoted the migration and invasion of head and neck cancer (HNC) cells with enhanced cancer stem cell-like properties and induction of epithelial-to-mesenchymal transition. Furthermore, neutralization of CCL11 activity reversed the aggressive phenotype of CAF-induced cancer cells. Confocal microscopy showed colocalization of CCL11 and CC chemokine receptor 3 (CCR3) on HNC cells. Moreover, immunohistochemical analysis of clinical samples from 104 patients with HNC showed that expression of CCL11 and CCR3 were significantly correlated with poor overall survival (*p* = 0.003 and 0.044, respectively). Collectively, CCL11 expressed on CAFs promotes HNC invasiveness, and neutralization of CCL11 reverses this effect. We propose that the CCL11/CCR3 signaling circuit is a potential target for optimizing therapeutic strategies against HNC.

## 1. Introduction

Head and neck cancer (HNC), the seventh most common cancer worldwide, represents a variety of malignant tumors of the upper aero-digestive tract [1]. More than 60% of patients with HNC present with locally advanced disease, characterized by widely extended tumor and/or multifocal growth with marked local invasion and/or regional lymph node metastases [2]. Emerging studies have shown that tumor cell invasion and migration are key events in tumor progression and metastasis [3]. The prognosis for patients with HNC is poor, even when contemporary multimodal treatments are used; thus, elucidating the underlying mechanisms that promote the invasiveness of cancer and finding a novel treatment strategy are crucial. 

Tumors are no longer regarded as a bulk of malignant cells, but rather as a complex system consisting of the tumor and its adjacent tumor microenvironment (TME), comprising various cell populations, including cancer-associated fibroblasts (CAFs), infiltrating immune cells, and the extracellular matrix. The TME plays an important role in cancer development and metastasis. A review of the literature showed that interactions between stromal cells and cancer cells, through various cytokines, chemokines, and other mediators, are associated with tumor invasiveness and treatment resistance [4]. CAFs are the key cell type in the TME and play essential roles in maintaining a favorable milieu and interacting with other stromal components, including immune cells and vessels, by secreting various inflammatory cytokines, chemokines, and other tumor-promoting factors [5,6,7]. Certain CAF-secreted cytokines may mediate immune escape by cross-talk with surrounding immune cells in TME, leading to cancer progression [8]. Moreover, CAFs may alter the immune status within tumors and affect response to current cancer immunotherapy [9,10]. 

Chemokine ligand 11 (CCL11), also known as eotaxin-1, is involved in cancer cell migration and invasion. Initially reported to recruit eosinophils to inflammatory sites, predominantly in allergy and asthma [11], it is also associated with psychiatric disorder, neurodegenerative disease, and infectious disease [12,13,14]. Recently, CCL11 was also found to be produced by fibroblast and associated with immune response [15,16,17]. Elevated levels of CCL11 have been reported in some cancer types, where they are associated with a poor prognosis [18,19,20,21,22]. Increased tissue and serum levels of CCL11 inhibit dendritic cell maturation and differentiation in the TME and promote antitumor immunity [23]. CCL11 increases MMP-3 expression via the CCR3-ERK pathway, thereby promoting cell invasion and migration in prostate cancer [24]. Conversely, the loss of CCL11 reduced tumor outgrowth [24,25,26]. The functional role of CCL11 in tumor–stromal signaling, particularly TME-mediated aggressiveness, remains controversial. We previously conducted a gene expression microarray to compare the expression profiles of genes encoding protein secreted by CAFs and normal fibroblasts (NFs) [27] and found that, in addition to IL33, another important mediator, CCL11, was overexpressed in CAFs compared to NFs. Therefore, in the present study, we aimed to investigate whether CCL11 expressed on CAFs influences HNC promotions and the underlying molecular mechanism. Furthermore, we also studied whether CC chemokine receptor 3 (CCR3), the main corresponding receptor of CCL11, is directly involved in signal transduction resulting in cancer progression. 

## 2. Materials and Methods

### 2.1. Cells and Reagents

The FaDu cell lines (ATCC, catalog number: HTB-43^TM^) were purchased from the American Type Culture Collection (Manassas, VA, USA). The TNPC-204 cell line (RRID: CVCL_A5WV) was purchased from the Bioresource Collection and Research Center (Hsinchu, Taiwan). Cells, including fibroblasts (CAFs and human gingival fibroblasts (NFs)) and HNC cell lines (FaDu and NPC 204), were cultured as previously described [27]. Tumor specimens and adjacent normal tissue were collected from patients with HNC (n = 10, see Table A1). CAFs and NFs were isolated from cancer cell-adjacent fibroblasts and non-cancer gingival tissues, located at least 3 cm away from the tumors obtained from patients with HNC undergoing excision. To determine aggressiveness organotypic forming efficiency, the organotypic with cancer part depth >200 μm (more than 15 layers) were counted with hematoxylin and eosin stain under light microscope (Leica Microsystems), and the aggressiveness organotypic forming efficiency (‰) = scored cancer layer depth/total plating HNC cells. Cells were treated for 6  h with recombinant CCL11 (rCCL11) protein with doses of 50 μg/mL (MBS142051, MYBioSource) and then, in a series of experiments, were pre-treated for 1  h with various enzymatic inhibitors of different signaling pathways: 50  μM FR180204 (ERK, inhibitor), 10 μM SB203580 (p38MAPK inhibitor), or 4 μM SP600125 (JNK inhibitor). This study was approved by the Institutional Review Board of the Kaohsiung Veterans General Hospital, Kaohsiung, Taiwan (protocol code KSVGH21-CT1-10, 2020/03/26 and KSVGH16-CT6-09, 2018/07/16).

### 2.2. Western Blot Analysis

The antibodies used are listed in Appendix A. Western blotting was performed following a previously described standard protocol [27]. The intensity ratio was showed in Appendix A and the uncropped Western blots showed in Appendix A.

### 2.3. RNA Extraction, Quantitative Real-Time PCR (qPCR), and ELISA

We extracted the total RNA from CAFs and NFs using TRIzol reagent (Invitrogen, Waltham, MA, USA); 1 μg of RNA was used for cDNA synthesis. To quantify gene expression, we performed qPCR using the StepOnePlus real-time PCR system (Applied Biosystems, Inc., Waltham, MA, USA). The primers used are listed in Appendix A. CAFs and NFs were seeded onto six-well plates at equal density and incubated for 24 h. Cells were then incubated for a further 24 h with serum-free Dulbecco’s modified Eagle’s medium (DMEM)–F12. CAFs and NFs supernatant CCL11concentrations were measured using the CCL11 (human) ELISA kit (R&D system) following the manufacturer’s protocol. For the sensitivity, the minimum detectable dose of human CCL11 was less than 5 pg/mL. For the specificity, the assay recognized natural and recombinant human CCL11. Cell culture supernatants were prepared at 50 ng/mL in calibrator diluent and were then assayed. Preparations of factors at 50 ng/mL in a mid-range recombinant human CCL11 control were assayed for interference. No significant cross-reactivity or interference was observed. The plates were developed using tetramethylbenzidine as a substrate (TMB, Sigma-Aldrich, St. Louis, MO, USA), and the absorbance was recorded using a microplate reader.

### 2.4. In Vitro Migration and Invasion Assays

In vitro transwell migration and invasion assays were performed. FaDu or NPC204 cells were suspended in culture medium containing 0.5% serum, and then plated in the upper chamber. In the invasion assay, a Matrigel–medium (1:2) mixture was applied onto the membrane of the upper chamber before seeding the cancer cells. Following 12 and 24 h of incubation for the migration and invasion assays, respectively, cells on the upper side of the filter were removed, and migratory or invasive cells were fixed in 4% formaldehyde before staining with the hematoxylin. The number of migrated/invaded cells was counted using light microscope. In the co-culture migration/invasion assays, CAFs or NFs were seeded on the outer chamber prior to the FaDu and NPC204 cells and then incubated for 24 h. The growth medium was then replaced with medium with or without an anti-CCL11 antibody. For CCL11-induced migration/invasion assays, rCCL11 protein (10 or 50 ng/mL; MBS142051, MYBioSource) was added to the lower chamber. Then, the assay was performed according to a standard migration/invasion protocol. Three independent experiments were conducted, and data were shown as mean ± SD. 

### 2.5. Organotypic Culture

Three-dimensional, organotypic culture was performed as follows [28]. A collagen I/Matrigel gel was prepared by mixing 8 volumes of collagen I/Matrigel (collagen I:Matrigel = 1:1) with 1 volume each of 10 × DMEM and fetal bovine serum (FBS) containing fibroblast cells (5 × 10^6^). The gel mixture was dispensed into a 12 mm Millicell insert (Millipore Corp., Bedford, MA, USA) inside a six-well culture plate. The mixture was allowed to set at 37 °C for 24 h before cancer cells (2 × 10^5^) were suspended in growth medium and seeded atop the gel mixture. After incubating for 24 h, cancer cells were exposed to air by removing the medium from the surface. The gel was then fed from underneath with complete media, which was changed daily. After 14 days, cultured tissue was fixed and embedded in paraffin for histological analysis.

### 2.6. Flow Cytometry

A single-cell suspension containing 1 × 10^6^ trypsinized cells and spheres was resuspended in 1 mL of phosphate buffered saline (PBS) and stained with fluorescent conjugated antibodies against CD24 (IM1428U, Beckman Coulter, Inc., Brea, CA, USA), CD44 (A32537, Beckman Coulter, Inc. CA, USA), CD10 (GTX78263, Genetex, Irvine, CA, USA), and GPR77 (MAB10254, R&D systems, Minneapolis, MN, USA) for 30 min at 4 °C. For detecting aldehyde dehydrogenase 1 (ALDH1) activity, the ALDEFLUOR assay kit (Stem Cell Technologies, Durham, NC, USA) was used. After labeling, cells were washed three times with PBS and subsequently stained with a fluorescein isothiocyanate (FITC)- or PE-labeled secondary antibody for 30 min in the dark. Cells were then analyzed on a flow cytometer after washing thrice with PBS. 

### 2.7. Confocal Microscopy

The colocalization of CCL11 and CCR3 was confirmed using confocal microscopy. The immunofluorescence of CCL11 overexpression FaDu and NPC204 cells was studied using CCL11 and CCR3 primary antibodies. The cells were incubated, respectively, with fluorescent secondary antibodies Alexa Fluor^®^ 488 or 647 (Thermo Fisher Scientific, Waltham, MA, USA), followed by a nuclear counterstaining with DAPI. Confocal images were visualized at 20× magnification using a TissueFAXS Plus System (TissueGnostics, Vienna, Austria) coupled onto a Zeiss^®^ Axio Imager Z2 microscope (Jena, Germany). Annotated cell borders were identified with the transmission channel.

### 2.8. siRNA Knockdown

To perform gene knockdown, 6 × 10^5^ cells were seeded in six-well plates, and the siRNA (50 nM) against CCL11 or control siRNA (Santa Cruz Biotechnology) was mixed with TransIT-TKO transfection reagent (Mirus, Madison, WI, USA) following the manufacturer’s protocol.

### 2.9. Lentivirus Overexpression of CCL11

CCL11 cDNA (Appendix A) was cloned into the mammalian expression vector pLenti-GIII-CMV-Luc-2A-Puro vector from Applied Biological Materials (ABM Inc., Richmond, BC, Canada). The viral supernatants were added into FaDu and NPC204 cells to construct stable CCL11 overexpression cell lines. Cells were further treated with puromycin (2 μg/mL) for one week to select stably transfected cells.

### 2.10. Tissue Microarrays and Immunohistochemistry

Clinical cancer samples and patient characteristics were obtained from 104 HNC patients. Tissue microarrays were constructed from formalin-fixed, paraffin-embedded (FFPE) tissue blocks. A 2-mm-diameter tissue core was punched out from the invasive front of each patient’s block and transplanted onto a recipient block. Immunohistochemical staining was performed. Briefly, after rehydration and microwave antigen retrieval, monoclonal antibodies against human CCL11 and CCR3 were applied, according to the manufacturer’s instructions. Staining intensity was scored as follows: no staining (score 0), faint staining (score 1), moderate staining (score 2), and strong staining (score 3). Total scores were calculated using the following formula: Total score = staining intensity score × percentage of staining distribution.

### 2.11. Statistical Analysis

Densitometry was used to measure the fold-change between and among conditions. The Mann–Whitney U test was used to test statistics when the number of samples/tests (N) was less than ten. A Kaplan–Meier survival analysis was used to test associations between CCL11 and CCR3 expression, and the clinical outcome. The univariable Cox proportional-hazards model was used to determine the potential prognostic factors for overall survival. Factors with *p* < 0.1 in the univariable analysis were included in the multivariable analysis. A Student’s *t* test was used to identify significant differences between experimental groups with n > 10 and a normal distribution profile. All statistical analyses were conducted on SPSS software v. 17.0 (SPSS, Inc., Chicago, IL, USA), with a value of *p* < 0.05 considered statistically significant. 

## 3. Results

### 3.1. Higher Levels of CCL11 Are Found in CAFs Than in NFs

A panel of CAFs and NFs was generated with specimens from 10 patients with HNC (see Table A1). CAF and NF samples were dissected, isolated, and prepared from HNC patients who underwent surgical resection. Morphological observations demonstrated that CAFs present with more cytoplasmic protrusions than the relatively uniform NFs (Figure 1a). A higher expression level of vimentin and α-SMA was found in CAFs than in NFs, determined using RT-PCR and Western blot analysis (Figure 1b). Flow cytometric analysis showed a marked increase in the activity of CD10 and GPR77 (Figure 1c), which were characterized as essential cell surface markers of a CAF subset correlated with chemo-resistance and poor survival [29,30]. Furthermore, we performed a gene expression microarray study to compare their expression profiles. We focused on a group of secreted proteins, including chemokine-related and/or cytokine-associated genes, which were significantly up- or down-regulated (≥4-fold) in CAFs compared to NFs. Among these, CCL11 expression was significantly higher in CAFs than in NFs (Figure 1d); this finding was also confirmed by RT-PCR and ELISA analysis (Figure 1e). Western blot analysis also showed CCL11 overexpression in CAF, compared with that in NF in cell lysates (Figure 1f). Moreover, administration of CAF-conditioned medium (CAF-CM) to FaDu and NPC204 cells, compared to NF-conditioned medium (NF-CM), induced a higher CCL11 expression (Figure 1g). Collectively, these results showed that CCL11 was overexpressed in CAFs, implicating that CCL11 was associated with CAF-induced cancer behaviors.

### 3.2. CAF-Induced CCL11 Increases the Ability of Migration and Invasion of HNC Cells and Promotes Epithelial-to-Mesenchymal Transition (EMT)

CCL11 overexpression in the CAF-culture medium suggests that CCL11 might be critical for CAF-induced cancer aggressiveness. FaDu and NPC204 cells cultured with medium containing CAF-induced CCL11 showed greater abilities of migration and invasion. Furthermore, migration and invasion assays using a CCL11-neutralizing antibody and rCCL11 were performed to confirm the functional role of CAF-mediated CCL11 (Figure 2a). The migration ability of FaDu and NPC204 cells increased in rCCL11-treated cancer cells (Figure 2a, upper panel); the results of the Transwell invasion assay showed a similar effect on the cell invasion ability of FaDu and NPC204 cells (Figure 2a, lower panel). Treating cells with an anti-CCL11 antibody neutralized its activity and attenuated the aggressive phenotype.

According to the results of the Transwell migration assay, migration ability increased 8–9-fold in FaDu and NPC204 cells cultured with CAF-CM, compared to those cultured in control medium. Increased invasiveness was also observed following CAF-CM treatment of FaDu and NPC204 cells. Furthermore, we used a three-dimensional organotypic culture system, a co-culture model mimicking the in vivo TME, to investigate the interactions, particularly invasion, between CAFs and cancer cells. Using the three-dimensional organotypic raft culture technique, we tested whether clinically isolated CAFs could induce a more aggressive phenotype and behavior in cancer cells. To determine whether the CAFs and NFs recapitulate the clinical features of original tumors, several experiments were conducted. To clarify whether CAFs promote the aggressiveness organotypic forming ability, CAF and NF cells were sorted out and co-cultured with cancer cells. The results showed that after co-culture with CAFs, the aggressiveness organotypic forming efficiency of CAFs was largely increased, while the NFs cells were unaltered, indicating that co-culture with CAFs may promote the invasive ability of CAFs. (see Table A1). FaDu and NPC204 cell infiltration was clearly observed in the section of the matrix layer containing embedded CAFs, supporting the role of CAFs in the aggressiveness of HNC. Among the four groups tested, the two groups with CAFs and NF+CCL11 showed a marked increase in invasiveness. Adding anti-CCL11 antibody reversed the invasion ability (Figure 2b). Overall, the above results showed that CAFs mediated a paracrine effect by means of CCL11 on cancer cells, thereby promoting cancer invasiveness. 

EMT encompasses dynamic changes in the cellular organization of epithelial-to-mesenchymal phenotypes, leading to increased cell migration and invasion; it is an important mechanism for cancer aggressiveness. We examined whether CAF-induced CCL11 promoted cancer cell aggressiveness by inducing EMT. First, we treated cells with a CAF or NF conditioned medium. Western blot analysis showed decreased expression of the epithelial marker E-cadherin and increased expression of the mesenchymal marker fibronectin; EMT regulators (Twist and Snail) were also overexpressed (Figure 2c). Furthermore, rCCL11 treatment also induced changes in EMT markers/regulators, resulting in a more mesenchymal phenotype. Matrix metalloproteinase-2 (MMP-2) and matrix metalloproteinase-9 (MMP-9) are capable of degrading type IV collagen, the most abundant component of the basement membrane. Degradation of the basement membrane is an essential step for the metastatic progression of most cancers. Treating FaDu and NPC204 cells with rCCL11 induced MMP-2 and MMP-9 expression. Collectively, these results support the hypothesis that rCCL11 plays a critical role in the promotion of cancer aggressiveness as a result of CAFs inducing an EMT-like conversion.

### 3.3. CCL11 Contributes to the Induction of Sphere Formation, Enhanced Cancer Stem Cell (CSC) Properties, and Drug Resistance

To understand the CCL11-mediated crosstalk between cancer cells and CAFs at the molecular level, we investigated if the paracrine effects of CCL11 effect the expression of the CSC-like properties of HNCs. The effects of CAF-induced CCL11 on the ability of sphere formation, the expression of CSC-like properties, and the expression of CSC-representative markers were first examined. A non-adhesive sphere culture system was used, and the result showed that HNC cell lines exposed to rCCL11 and CAFs-conditioned medium initially generated immature, floating spheroids in the first 3–5 days, which further gradually transformed into well-formed spheres at day 7. However, control cells yielded only an irregular small cell mass without the spheroid appearance (Figure 3a). Flow cytometric analysis showed significantly higher CD44/CD24 and CD133 in HNC cells exposed to rCCL11, compared with control HNC cells (*p* < 0.05) (Figure 3b). Moreover, the fluorescence level of ALDH1 obtained through FACS analysis was higher following treatment with rCCL11 in both HNC cells (Figure 3c). Western blot analysis showed that other CSC-representative markers, including OCT4, Nanog, and Sox-2, were also overexpressed in HNC cells exposed to CCL11 compared to control HNC cells. Drug resistance-related gene expression was also tested in HNC cells. Western blot analysis showed that CCL11 was also involved in *ABCG-2* and *MDR-1* upregulation (Figure 3d). Chemoresistance treated with Cisplatin was also significantly enhanced in both FaDu and NPC204 cells exposed to rCCL11 (Figure 3e). Collectively, these results indicate that cells exposed to CCL11 through paracrine signaling enhance representative CSC properties and drug resistance.

### 3.4. Expression of CCL11 and Its Receptor, CCR3, in HNC Cells Is Closely Associated with Clinical Patient Survival

CCR3 is reported to be a specific receptor for CCL11; the interaction of CCL11/CCR3 directly induced cell survival and proliferation [24,25]. Confocal microscopy analysis showed the cellular co-localization of CCL11 and CCR3 in HNC tumor cells (Figure 4a). This polarized co-localization of CCL11/CCR3 signaling suggests evidence of a chemotactic effect that might be important for sensing the environment and guiding migration in HNC cells. Collectively, these findings supported our hypothesis that CCL11 would stimulate the chemotactic migration of HNC cells. 

Cloned CCL11-overexpressed FaDu and NPC204 cells were first established. Western blot analysis showed that the CCL11-overexpressed FaDu and NPC204 cells presented higher expression level of CCL11, CCR3, MMP2, and MMP9. These proteins were reversed by treatment of the CCL11 siRNA and CCR3 antibody (Figure 4b). To identify the downstream pathway involved in CCL11/CCR3 signaling, we examined the phosphorylation level of p38 MAPK, ERK, and JNK. Western blot analysis showed high expression of p38 MAPK and ERK, rather than JNK, in CCL11-overexpressed FaDu and NPC204 cells, which were further suppressed after treatment of their individual inhibitors (Figure 4c). This result indicates that the CCL11/CCR3 downstream pathway was involved in p38 MAPK and ERK, resulting in an enhanced ability of invasion and migration. 

Immunohistochemical staining of CCL11 was observed in the putative area where CAFs and tumor from a representative case delineated the same area, and distribution of CCR3 investigated in the same area showed that both expressions coexisted (Figure 4d). Furthermore, we validated the role of CCL11 and CCR3 expression on the prognosis of patients with HNC using immunohistochemistry on tissue microarrays of clinical samples. The patient characteristics and clinical parameters of the 104 patients whose samples were used in this study are shown in Appendix A, based on the expression of CCL11 and CCR3. In survival analysis, higher CCL11 expression is associated with lower 5-year overall survival (Figure 4e, 62.4% vs. 79.1%, *p* = 0.003). For CCR3, higher expression was also significantly associated with lower 5-year overall survival (Figure 4e, 62.3% vs. 74.4%, *p* = 0.044). To identify independent prognostic factors for overall survival, we conducted univariable and multivariable analysis using the Cox proportional-hazards model. The results are shown in Appendix A. In univariable analysis for possible prognostic factors, T stage (T1-2 vs. T3-4), expression of CCR3, expression of CCL11, and co-expression of CCL11 and CCR3 were significantly associated with overall survival in patients with HNC. In multivariable analysis, only T stage, expression of CCL, and co-expression of CCL11 and CCR3 were independent prognostic factors for overall survival. Thus, the most important prognostic factors were T stage and expression of CCL11. When patients had both T3-4 disease and overexpression of CCL11, the 5-year survival was only 46.2%; significantly lower than the other three subgroups (*p* < 0.001, Appendix A). 

## 4. Discussion

Cancer cells inhabit a complex microenvironment in vivo, comprising non-cancer cells and tumor stroma, each of which contributes to cancer development and progression. Modulation of the TME, apart from altering the hemostasis cascade, is regarded as being closely associated with cancer behaviors. The presence of active CAFs around cancer cell nests is associated with poorer survival in solid cancers, such as cancers of the esophagus, prostate, colon, and breast [31,32,33,34]. However, the underlying molecular mechanism and role of CAFs in TME remain unknown. CCL11 has been reported as a diagnostic or prognostic marker in cancers [19,22,25,26,35]. However, other than studies on allergic responses or immunological function involving CCL11, there is a lack of studies showing its role in other diseases, especially HNC. Here, we provide a clear step-by-step demonstration of CCL11 as a critical mediator in CAF-induced cancer invasiveness. 

We first isolated and cultured the NFs and CAFs from the tissue of patients receiving head and neck surgery at our institute, because using patient-derived fibroblasts may mimic clinical situations. CAFs in the TME are thought to be transformed from NFs. We observed a progressive transition from NFs to CAFs, showing functional and morphological alterations between them. As reported previously [29], CAFs in primary culture displayed a higher expression of CD10 and GPR77 than NFs, suggesting that these molecules help the differentiation of CAFs from NFs. Moreover, CAFs displayed more peculiar cytoplasmic protrusions than NFs, and they also displayed an increased ability to induce HNC cells to become more aggressive compared to NFs. Induction of invasion was markedly increased by CAF-induced CCL11 via the paracrine effect. In this study, we used an organotypic culture system to investigate the interactions between CAFs and cancer cells. As seen in Figure 2, FaDu and NPC204 cell infiltration was clearly observed in the section of the matrix layer containing embedded CAFs, supporting the role of CAFs and their secretions in enhancing HNC aggressiveness. We performed a gene expression microarray to examine the differences in the expression profiles of cytokines and chemokines between CAFs and NFs, showing a marked increase in CCL11 expression in CAFs compared to NFs.

This intriguing and novel finding was validated in the present study by showing that the functional CAF-induced CCL11 plays a driving role in cancer behaviors via induction of EMT and enhancing CSC-like properties. Treatment with rCCL11 induced the EMT phenotype in our cells, suggesting that CCL11 is a novel mediator in CAF-induced cancer aggressiveness through the induction of EMT. Furthermore, the CSC representative marker expression, sphere-forming ability, and chemo-resistance with drug-resistant gene upregulation (*ABCG2* and *MDR-1*) was observed in HNC cells exposed to rCCL11. 

In our result, we verified the existence of the main corresponding receptor of CCL11, CCR3, that is directly involved in signaling transduction and cancer progression by confocal microscopy and immunohistochemistry. CAF-induced CCL11 chemotactically migrated and interacted with CCR3 situated on HNC cells, reflecting the existence of a CCL11/CCR3 signaling axis or circuit. We also validated that the CCL11/CCR3 downstream pathway strongly associated with p38 MAPK and ERK signaling cascade. Furthermore, to examine the clinical significance of the expression of CCL11 and CCR3, we immunohistochemically analyzed tissue samples from 104 representative patients with HNC for whom clinical information was available. We constructed a tissue microarray for analysis of a relatively large histopathological sample size. It showed that individual expression of CCL11 or CCR3, as well as co-expression of both CCL11 and CCR3, was significantly correlated with poor overall survival. These results suggested that CCL11 might be a potential prognostic biomarker in HNC. 

The tumor stroma offers an attractive target for cancer therapeutics and may provide a new route for treatment intervention. The results of our present study directly link the most two important hallmark events of cancer behavior, invasion and microenvironment. We demonstrated the role of CCL11 in the CAF-promoted aggressiveness of cancer cells, as well as the mechanisms involved in the interactions between cancer cells and fibroblasts. Through CCL11, CAFs interact and crosstalk with cancer cells, eventually leading to tumor progression. Additionally, we verified that CCL11 expression from surrounding CAFs directly affects the tumor behavior and promotes its aggressiveness by targeting CCR3. This effect may subsequently induce self-secretion of CCL11 from the tumor, leading to a vicious cycle of spontaneous CCL11/CCR3 circuit activity, which is now shown to be correlated closely with invasive behavior and clinical outcomes in patients with HNC. This kind of complex crosstalk with molecular communications proceeds and conducts via both paracrine and autocrine signaling between CAFs and the tumor. The diagrammatic illustrations (Figure 5) of three sequential steps demonstrate our proposal to explain the major mechanism of how CAF-induced CCL11/CCR3 signaling in TME contributes to cancer behavior.

## 5. Conclusions

Our results provide an insight into the mechanisms underlying the interaction between CAF and HNC cells via the paracrine effect of CAF-induced CCL11. Further work on the genome sequencing of CAFs and the patient-derived tumor explant model is warranted on the basis of clinical translation. For precision medicine-based clinical practice, modulating the TME by targeting the CCL11/CCR3 signaling circuit may attenuate the aggressiveness of cancer cells and offer a novel therapeutic strategy to improve the prognosis in patients with HNC.

## Figures and Tables

**Figure 1 cancers-14-03141-f001:**
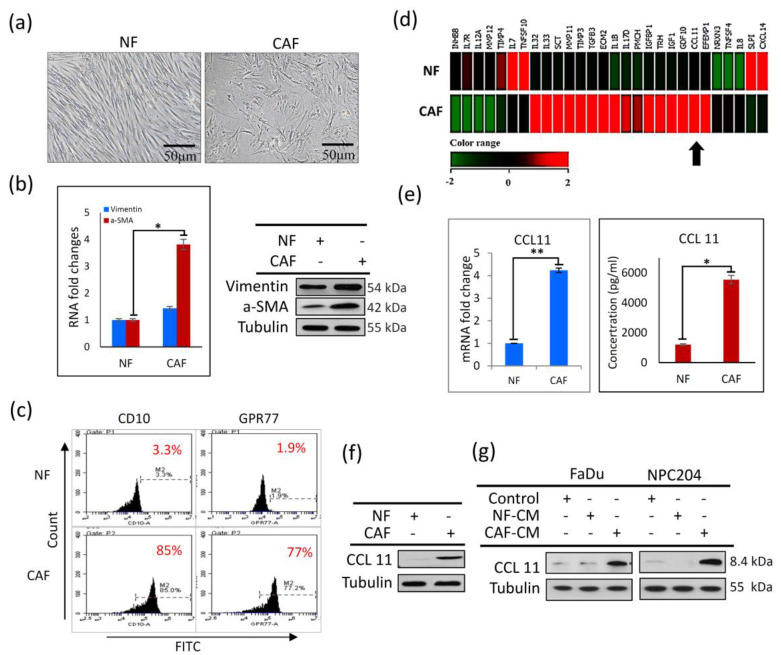
Characterization of CAFs and NFs obtained from clinical surgical tissues from patients with head and neck cancer (HNC) (**a**) Morphological comparisons between CAFs and NFs from a representative HNC case showed that CAFs (right panel) consisted of more cytoplasmic protrusions than NFs (left panel). Photographs were captured at 40× magnification. (**b**) Quantitative PCR (left panel) of the culture medium showed a significantly higher expression of vimentin and α-SMA in CAFs than in NFs. Western blot analysis (right panel) also demonstrated that levels of vimentin and α-SMA were significantly higher in CAFs than in NFs. (**c**) Flow cytometric analysis of the cell surface markers, CD10 and GPR77, showed a marked increase in their expression in CAFs compared to NFs. (**d**) A heat map of the gene microarray of NFs and CAFs showed that there were several differences in the expression profile of the secreted genes. The arrow indicates a marked discrepancy of the relative mRNA levels of CCL11 in CAF compared with NF. (**e**) The RT-PCR (left panel) and ELISA (right panel) analysis showed an increased expression of CCL11 in CAFs compared with that in NFs. (**f**) Western blot analysis showed that the protein level of CCL11 was higher in CAFs than in NFs in cell lysates. (**g**) Western blot analysis showed a higher CCL11 expression in CAF-CM compared to NF-CM. The asterisk indicated a significant difference (*: *p* < 0.05; **: *p* < 0.01) between experimental and control groups.

**Figure 2 cancers-14-03141-f002:**
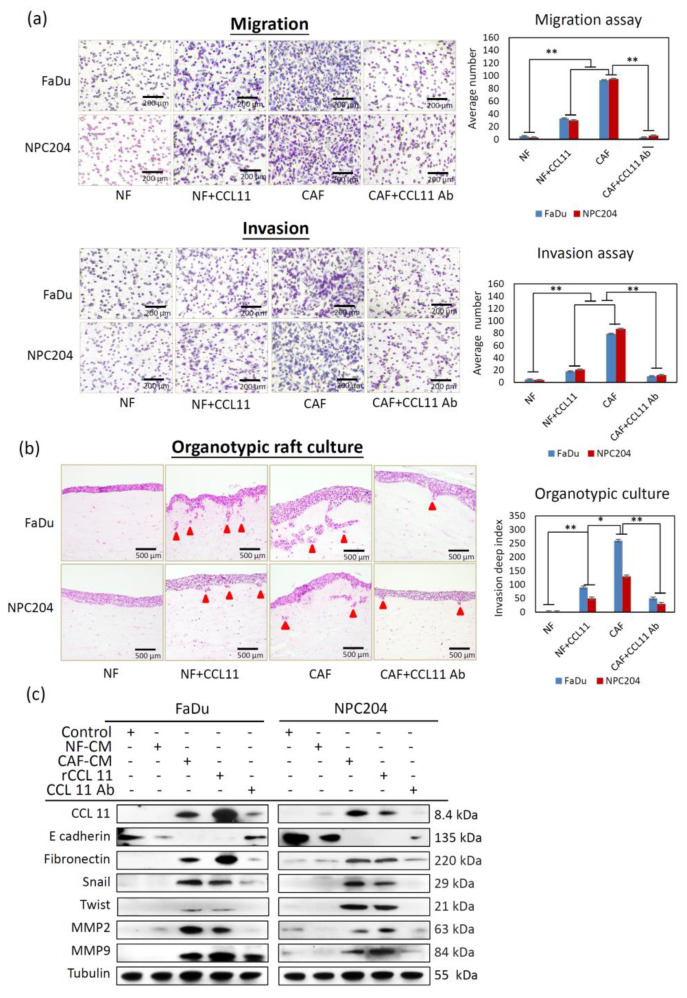
CCL11 produced by CAFs causes increased migration and invasion, and the EMT of HNC cells. (**a**) Comparative analysis of the migration and invasion of HNC cells associated with CCL11. Four test groups were classified for comparative analysis of migration and invasive abilities. FaDu and NPC204 cells cultured with medium containing CAF-induced CCL11 presented greater abilities of migration and invasion, with a statistically significant difference, than three other groups: NF, NF with CCL11, and CAFs treated with CCL11 antibody. The asterisk indicates a significant difference (*: *p* < 0.05; **: *p* < 0.01) (**b**) Comparative photographs of the infiltrating behavior of FaDu and NPC204 cells in an organotypic culture in four groups seeded onto a mixture layer containing NFs or CAFs with CCL11 or CCL11 antibody. The arrow(s) indicate infiltration buds from the HNC cells seeded above. (**c**) Representative blots of the EMT-associated markers in FaDu and NPC204 cells, as observed upon Western blotting analysis in five groups, showed that treatment with CAF-conditioned medium or the application of rCCL11 decreased the expression of epithelial-type markers (E-cadherin), and increased the expression of mesenchymal-type markers (fibronectin) and EMT regulators (Snail and Twist). In addition, increased expression of invasion-related MMP2 and MMP9 was also seen in those two groups, compared with other groups. The asterisk indicates a significant difference (*p* < 0.05) between experimental and control groups. Results are expressed as mean ± SD.

**Figure 3 cancers-14-03141-f003:**
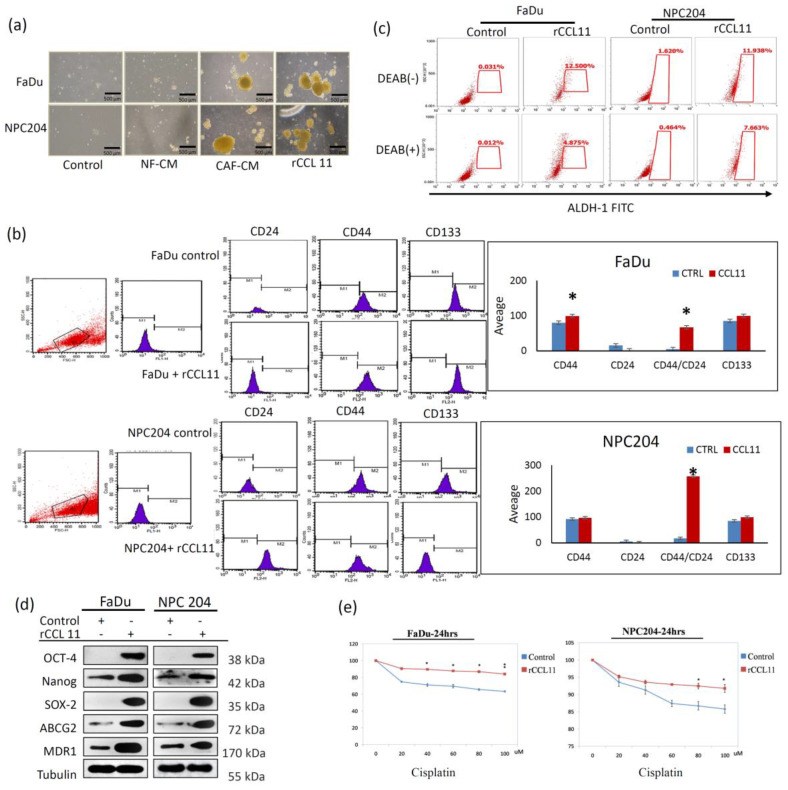
Comparative analysis of induction of CSC properties and drug resistance in HNC cells associated with CCL11. (**a**) Four groups were classified for comparative analysis of the ability of sphere formation. Increased ability of sphere formation in two test groups of HNC cells exposed to a CAF medium and the group with treatment of rCCL11 was noted. (**b**) Flow cytometric analysis showed a significant increase in CD44 and CD44/CD24, as well as in CD133 in HNC cells exposed to rCCL11, compared to control HNC cells (*p* < 0.05). (**c**) Flow cytometric analysis showed a marked increase in ALDH-1 activity in HNC cells exposed to rCCL11 compared to control HNC cells. (**d**) Western blot analysis showed that CSC-representative markers, Oct-4, Nanog, and Sox-2, were also overexpressed in addition to the increased expression of two important drug resistance genes, *ABCG-2* and *MDR-1*, in HNC cells exposed to rCCL11. (**e**) Treatment with Cisplatin at 24 h showed a significant increase in chemoresistance in both FaDu and NPC204 cells exposed to rCCL11 compared with control HNC cells. The asterisk indicates a significant difference (*: *p* < 0.05; **: *p* < 0.01) between experimental and control groups.

**Figure 4 cancers-14-03141-f004:**
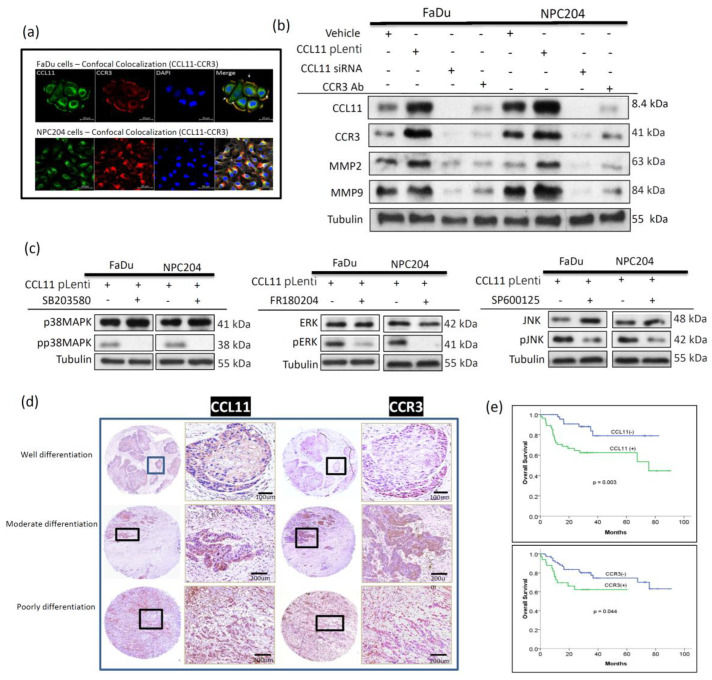
CCL11 and CCR3 expression with associated signal pathway in HNC cell lines and their correlation to clinical outcomes in 104 HNC patients. (**a**) Confocal microscopic images showed CCL11 (green) localized to both cell and nuclear membranes, while CCR3 (red) localized only to the cell membrane in FaDu cells; CCL11 and CCR3 co-localized at the cell membrane (yellow). In NPC204 cells, CCL11 (green) and CCR3 (red) were found to co-localize at protrusions polarized to the cells (yellow). (**b**) Using the crisp technique, higher expression of CCR3, MMP2, and MMP3 was found in over-expressed CCL11 cloned-FaDu and NPC204 cells. Cloned CCL11-overexpressed cells were abolished by adding eotaxin siRNA or CCR3 antibody, which reversed the expression of CCR3 and invasion-related MMP2 and MMP9. (**c**) Higher phosphorylation levels of p38 MAPK and ERK were found in cloned CCL11-overexpressed FaDu and NPC204 cells and were reversed by treatment of the p38 MAPK inhibitor (SB203580) and ERK inhibitor (FR180204), respectively. The phosphorylation level of JNK was kept in low condition before and after treatment of the JNK inhibitor (SP600125). (**d**) Photomicrographs of immunohistochemical staining from tissue microarray showing CCL11 and CCR3 expression in three different representative groups of HNC patients (magnification, ×200). (**e**) Kaplan–Meier survival analysis of patients showed that overexpression of CCL11 and CCR3 were statistically associated with poor overall survival).

**Figure 5 cancers-14-03141-f005:**
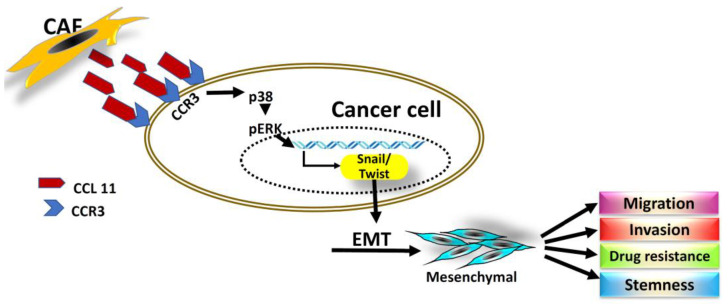
The diagrammatic illustration demonstrates the major mechanism that CAFs secreting CCL11 promotes HNC cell migration and invasion and induces properties of drug resistance and stemness, shown as follows. CAFs in TME secret CCL11 binding to the CCR3 receptors on HNC cells via the paracrine effect. The signal induces overexpression of transcriptional factors, such as Snail and Twist, which regulate EMT and are also responsible for self-induction of CCL11 in an autocrine fashion. As a result, CCL11, via paracrine or autocrine signaling when targeting CCR3 receptors, play a functional role in the induction of EMT and CSC properties, for further tumor progression.

## Data Availability

The data in the study are available on request from the corresponding author.

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
