# Peer review of "Cancer-Associated Fibroblasts Promote Tumor Aggressiveness in Head and Neck Cancer through Chemokine Ligand 11 and C-C Motif Chemokine Receptor 3 Signaling Circuit"

_cancers, 2022, doi:10.3390/cancers14133141_

Round 1
Reviewer 1 Report
This manuscript by Huang et al. presents data that collectively suggest the CCL11-CCR3 mediated crosstalk of cancer associated fibroblasts and tumor cells in HNC contributes to the poor outcomes. The authors demonstrate that the CAFs in the HNC tumors secrete high levels of CCL11 compared to normal fibroblasts. The authors identify that the CCL11 secreted from CAFs promotes the aggressiveness of tumor cells by enhancing the migratory and invasive properties. Additionally, they demonstrate increased expression of cancer stem cell properties and induction of the epithelial-to-mesenchymal transition in tumor cells treated with CAF-conditioned media and rCCL11. The data presented are clear and of interest. However, few concerns need to be addressed.
Major comments:
1- From Fig. 2b, it is not clear that the anti-CCL11 targets the CCL11 secreted by CAFs, or the tumor cells itself (Fig.3a & b) to inhibit the infiltration of tumor cells. Repeating this experiment with CCL11 siRNA treated tumor cells will clarify on the role of CAFs in this study.
2- For Fig.4a, Do the authors treat the cell lines with CCL11 or CAF-CMs? Based on the microscopy images, the author stated ‘This polarized co-localization of CCL11/CCR3 signaling suggests evidence of chemotactic effect that might be important for sensing the environment and guiding migration in HNC cells. Collectively, these findings supported our hypothesis that CCL11 would stimulate the chemotactic migration of HNC cells.’ The reviewer would like to know whether the CCL11 secreted by the tumor cells is also involved in promoting the migration of tumor cells.
3- The tubulin bands presented in Fig.1g do not seem to be matching with the ones shown in the supplement files; the reviewer suggests a repeat of the experiment. However, if revising Fig.1g is not possible, at a minimum figure from another independent experiment should be provided.
4- There are few references missing in the manuscript. Specifically, for
‘Flow cytometric analysis showed a marked increase in the activity of CD10 and GPR77 (Figure 1c), which were characterized as essential cell surface markers of a CAF subset correlated with chemo-resistance and poor survival.’
and
‘CCR3 was reported to be a specific receptor for CCL11; the interaction of CCL11/CCR3 directly induced cell survival and proliferation.’
Minor comments:
5- All the figure legends describe the results derived from the figures and do not mention the description used for the acronyms.
6- In Fig. 2a there is inconsistency between HGF and NF labeling for the pictures.
7- In Fig. 3b the Y-axis do not mention the parameter associated with the average. Also, there is a typo in the spelling.
8- Line 375-JUK for JNK; Line 443-p38 MARK for p38 MAPK
9- The statement made in Lines 442-446 is unclear and needs to be redrafted.
Reviewer 2 Report
The current study by Huang et al. demonstrates that targeting CCL11-CCR3 signaling could be a promising therapeutic strategy for patients with aggressive HNSCC. Their research sheds light on the mechanisms underlying the interaction between CAFs and HNC cells via the paracrine effect of CAF-induced CCL11 on HNC cells, which conjugates directly to its corresponding receptor, CCR3. The findings will improve prognosis in HNC patients in the future and promote precision medicine-based clinical practice. The manuscript is very well written, and the findings are very solid, with a great experimental setup. There are no major changes or modifications required, but I do have some suggestions that may help to improve the manuscript's quality: -
- It would be fantastic if the author could perform unbiased bulk RNA sequencing from CAFs and NFs, which would provide novel and complete information.
- It would be interesting to go over the correlation with the human data that is already available in greater detail (In-Silco data analysis).
- Please include the ELISA kit's sensitivity and specificity.
- Authors should comment on their study's statistical power.
- Conclusions should be expanded to better explain future perspectives based on the findings of this study.
- It would be fantastic if the author could replicate the results in a tumor explant model where tumor fragments are evaluated ex-vivo and tumor heterogeneity and tumor–stromal interactions are taken into account.
Reviewer 3 Report
Dear authors,
Overall work here, I believe the authors have done an exemplary job in preparing this manuscript. The level of scientific rigor is apparent, and the attention to detail in regard to every aspect of the replication is appreciated.
This study is good and needs to be improved for our audience. It contains more lack of literature evidence which is not possible to complete the story here and validated methodology and need to be improved further to build this study strong enough to publish.
Reviewer 4 Report
In this manuscript, the authors have discovered overexpression of CCL11 from cancer-associated fibroblasts (CAFs) compared with normal fibroblasts (NFs) isolated from their head and neck cancer (HNC) patients. The authors further identified that overexpression of CCL11 increased migration and invasion ability of HNC cells, promoted EMT process, induced sphere formation, enhanced cancer stem cell properties and drug resistance to cisplatin. In addition, they evaluated the association of expression of CCL11 and its receptor of CCR3 to survival rate in their HNC patients. Overall, they suggested that targeting the CCL11/CCR3 signaling pathway might improve the prognosis in HNC patients in clinic. However, there are some remaining questions to be answered:
1, Have the authors evaluated the CCR3 expression based on microarray data? Does the expression of CCR3 correlated with CCL11?
2, In Fig. 2, although HNC cells treated with CCL11 and CAF showed increased migration and invasion ability, HNC cells treated with CAFs have more dramatic effect compared to HNC cells with CCL11. Could the authors explain the possible reasons? Does the amount of exogenous CCL11 is less compared with CCL11 produced by CAFs? Does the activity of exogenous CCL11 is lower? Or is it possible that other factors produced by CAFs contribute to the effect?
3, In Fig 3, have the authors evaluated the proliferation rate of HNC cells treated with CAFs? Does coculture of HNC cells and CAFs or cells treated with CAF-CM promote cell growth compared to NFs? Have the authors examined the drug resistance within HNC cells treated with CAFs or CAF-CM besides recombinant CCL11?
4, In Fig 4b, the expression levels of endogenous CCL11 were decreased with CCR3 Ab and vice versa. Considering CCR3 is only the receptor of CCL11, could the authors discuss why their expression level were affected by each other?
5, Is there any commercial CCL11 inhibitors? It would be ideal to test the effect if there are specific CCL11 inhibitors available.
6, Have the authors evaluate the correlation between the expression of CCL11 and CCR3 and overall survival based on external databases? Does higher expression of CCL11 and CCR3 correlated with decreased survival rate in HNC patients?
7, Line 105, dot before [27] should be moved after [27].
8, Fig 2a, HGF, HGF+CCL11 under images doesn’t match to NF, NF+CCL11 under histograms and figure legends.
Round 2
Reviewer 1 Report
Huang et. al. has addressed all the issues identified by the reviewer.
Author Response
Reviewer 1
Comments and Suggestions for Authors:
Huang et. al. has addressed all the issues identified by the reviewer.
Reply: We appreciate the reviewer’s thorough review.